# OpenReview forum: "SyncMind: Measuring Agent Out-of-Sync Recovery in Collaborative Software Engineering"
_ICML.cc/2025/Conference — ICML 2025 poster_

### Official Review · Reviewer_VWuA · 2025-03-16

**Overall Recommendation:** 4

**Summary:**

This paper introduces SyncMind, a framework designed to analyze and measure how AI agents (specifically LLMs) handle “out-of-sync” challenges in collaborative software engineering (CSE). The out-of-sync problem arises when multiple collaborators modify a shared codebase at different times, causing one collaborator’s local understanding (the “belief state”) to diverge from the codebase’s current state. To systematically study this issue, the authors create SyncBench, a large-scale benchmark of 24,332 real-world out-of-sync scenarios derived from 21 GitHub repositories with executable test environments. They evaluate various LLM-based coding agents on SyncBench, measuring dimensions such as out-of-sync recovery success, collaboration willingness, and resource awareness. Results show substantial performance gaps among different agents, limited collaboration tendencies, and insufficient resource-awareness, thereby highlighting challenges and opportunities in building effective AI collaborators for real-world software engineering.

**Claims And Evidence:**

Yes

**Essential References Not Discussed:**

NA

**Experimental Designs Or Analyses:**

Yes, the main concern is the limited scope of the proposed benchmark.

**Methods And Evaluation Criteria:**

Yes, they make sense

**Other Comments Or Suggestions:**

- Expand the dataset to include a wider variety of programming languages and domains (e.g., front-end frameworks, data science projects) for greater generality.
- Provide a breakdown of the most common out-of-sync failure modes encountered by the agents, which would help researchers target specific weaknesses.
- Offer granular collaboration analysis, such as deeper insights into how agents collaborate (e.g., frequency of clarifying questions, quality of the requests, how they handle conflicting suggestions), not just whether they do.
- Consider modeling different “budget profiles” (e.g., small open-source team vs. large corporate environment) and see how resource constraints might change agent behavior or performance.
- In future work, consider user studies that test how human developers respond when the agent is out-of-sync, and whether certain agent behaviors foster more trust or better synergy.
- In terms of the wirting and typos, the authors seem to have used a lot of latex commands to narrow the spacing, e.g., at the end of P4,6,7,8, the texts on the left and right sides are not on the same level.

Overall, SyncMind and SyncBench bridge a crucial gap in AI-driven software engineering, connecting multi-agent collaboration literature with practical, real-world coding environments. By doing so, they pave the way for more robust, resource-aware, and interactive AI collaborators.

**Other Strengths And Weaknesses:**

### Summary of Strengths
- The paper addresses a critical yet under-explored problem—how AI agents can detect and recover from out-of-sync states in collaborative coding, where codebases evolve dynamically.
- This paper propose a useful bemchmark --- SyncBench, which provides 24,332 test instances from real-world GitHub repositories, ensuring the benchmark reflects realistic CSE scenarios.
- The authors measure not just accuracy or success rate, but also collaboration willingness, communication effectiveness, and resource-awareness—factors critical to real-world teamwork. The multi-Dimensional evaluation enhance the soundness of this paper.
- Empirical results demonstrate that collaboration can improve success rates and that agents often lack effective resource-allocation strategies. These findings give concrete directions for future research and development.
- The paper is well-orginizied and well-written, especially the figures are clear and well-colored, which is easily understood,

### Summary of Weaknesses
- While 21 GitHub repositories is a good start, there still remains the concern on the limited scope of repositories, as the range of languages, frameworks, and complexity might still not capture the full breadth of real-world software projects.
- Although the paper highlights resource-awareness as a key dimension, the specifics of how computational/time expenses are measured and how an agent’s decisions are scored or penalized for resource overuse could be more transparent.
- While the results highlight performance gaps, it is unclear which error types or failure modes are most frequent (e.g., syntax errors, semantic misunderstandings, version conflict misunderstandings).
- The paper focuses on LLM-based agents. It remains to be seen whether the SyncMind framework readily extends to other types of AI collaborators or more specialized models (e.g., symbolic reasoning systems).

**Questions For Authors:**

- Could you elaborate on the types of resource constraints (time, compute, monetary cost) used in your experiments, and how they are enforced or simulated in SyncBench?
- Do you categorize different types of failures (e.g., syntax vs. semantic vs. version conflict) to provide a more granular analysis of where agents fail?
- Have you tested or considered repositories in multiple programming languages? If so, does SyncBench handle multi-language codebases effectively?
- Do you envision extending SyncBench to incorporate real-time concurrency, or human-in-the-loop feedback (e.g., partial merges, code reviews), to better mirror complex team workflows?

**Relation To Broader Scientific Literature:**

- Multi-Agent Collaboration & Version Control: The out-of-sync challenge aligns with established work on multi-agent systems that emphasizes dynamic belief updating (e.g., partial observability in robotics, multi-agent planning in dynamic environments). SyncMind specifically tailors this perspective to software engineering, building on version control and conflict detection tools (e.g., Git) but extending them to handle semantic misalignments.
- LLM-Based Code Generation: Recent research has focused on coding assistants (e.g., Copilot, ChatGPT, Claude), typically evaluated in static, single-user scenarios. SyncMind addresses a gap by introducing a dynamic, multi-developer context, pushing the boundaries of code generation research to consider synchronization, environment changes, and real-time collaboration.
- Human-in-the-Loop Systems: The paper’s emphasis on agent-human collaboration and resource-awareness echoes broader literature in interactive machine learning, which examines how AI systems can adaptively seek help from human collaborators. SyncMind provides an empirical benchmark to measure how effectively LLMs can engage in such interactive problem-solving.
- Resource-Efficient AI: With increasing concern over the computational and monetary costs of large models, this work contributes by examining how LLM-based agents allocate resources (e.g., repeated code compilations, test runs, or queries for assistance). The findings align with the growing literature on green AI and efficient inference strategies.

**Theoretical Claims:**

Yes. They are correct

---

> ### Author Rebuttal · Authors · 2025-04-01
>
> Thank you for your valuable comments. We are honored that you find our work to be well-organized and well-written. Your kind suggestions, such as generalizability, granular analysis, budget profiles, and user studies also provide constructive insights that we would like to take into consideration in our revision.
>
> Allow us to first reply to most of your mentioned Weakness and Other Comments and Suggestions, and incorporate our responses to remaining Weakness and Other Comments and Suggestions into our question responses. Huge thanks for all your insightful comments and questions.
>
> 1. Our discussions and goals for future improvements (Sec A) resonate greatly with your valuable insights. In this work, we aim to provide the foundation for general agent out-of-sync in collaborative scenarios to benefit both human-AI and multi-agent collaborations. Constructing Caller and Callee to reflect real-world complexity hierarchy, experimental comparisons between Caller and Callee reveal the influence of task complexity on agents’ performance and behaviors. We then extend our discussion to real-world nuances and future improvements in Sec A, such as designing fine-grained complexity hierarchy and out-of-sync categorization, expanding SyncMind and SyncBench to other languages, collaboration systems, models, and domains, etc. We will also enrich our experiments with human-in-the-loop experiments and user studies (particularly aim to provide meaningful insights for human-AI collaborations), complex and adaptive resource metrics, granular analysis and pattern summarization, etc.
>
> 2. Reply to Questions
>
> (1) Our resource-aware framework (Sec 2.3) includes cost and time as two dimensions of resources, where time is quantified as the number of turns, and cost is measured by an initial budget (i.e., agent’s discretionary budget at the beginning of the task), collaborator assistance cost (i.e., to quantify the time and cost consumed by the collaborator to gather information and assist the agent along with the time and LLM call cost of the agent), and solution validation cost (i.e., to quantify the time and cost of the agent to arrive at current state and propose the solution, along with the time and cost, e.g., build testing env and execute tests, to evaluate agent’s solution on five metrics). Therefore, the time/cost is an estimated amount of all types of time/costs, such as computation time/costs, build and execution time/costs, LLM call time/costs, employment time/cost, etc. They are enforced on SyncBench samples throughout agents' out-of-sync recovery by including all initial resource availability and cost in system prompt, and prompting resource consumption and remaining availability along with agents’ task evolution (Sec D-E). Our experiments further explore the effects of different resources’ availability, with deeper analysis on LLM agents’ resource awareness and strategic utilization (Sec 4.7, C.5).
>
> (2) We totally agree to extend our failure analysis with more fine-grained categorization and summarization. We categorize your mentioned ‘failures’ as general types of out-of-sync causes (Fig 1), and agents’ out-of-sync recovery failures are further categorized based on our evaluation metrics, e.g., file localization failures, function localization failures, solution failures, etc., with detailed analysis and discussions (Sec 3.4, 4, C). We also perform in-depth analysis (Sec C) on LLM agents’ collaboration initiative, communication capabilities, action planning, and recovery strategies, with fine-grained elaboration on multiple aspects that affect agents’ performance based on five metrics (Sec 3.4). We will also extend our failure analysis to include a more granular analysis based on failure cases and multi-level categorization (Sec C, E.2).
>
> (3) Yes, all of our 21 source repositories involve multiple programming languages (mainly Python) and can be properly handled by SyncBench. Replying to both your question and your earlier mentioned weakness: Although more than many existing SE benchmarks, e.g., SWE-Bench that leverages 12 popular github repositories, we agree that 21 repositories may still limit generalizability. Defining agent out-of-sync to be language-agnostic, we consider the differences among diverse languages, hoping to focus on one language at each time with sufficient data to better evaluate and improve LLM agents’ out-of-sync recovery abilities. Making our methods (adaptable to different repositories and languages) open-source to help the community expand and customize SyncBench, we will also enrich SyncBench with repositories with different primary languages, complexity, and domains.
>
> (4) Yes, we will definitely include them in future work, along with our future improvements in Sec A, to reveal deeper insights for real-world complexity and human-AI collaborations. We will also embrace your other constructive suggestions, such as granular analysis, failure modes, budget profiles, user studies, etc.

---

### Official Review · Reviewer_4zz5 · 2025-03-17

**Overall Recommendation:** 3

**Summary:**

This paper introduces SyncMind, a framework that systematically defines the ``out-of-sync'' problem in collaborative software engineering in an agentic context, where an agent's belief state ($B_k$) diverges from the actual world state ($S_k$). Based on this framework, the authors create SyncBench, a benchmark featuring 24,332 instances of agent out-of-sync scenarios derived from 21 popular GitHub repositories with executable verification tests. The benchmark includes two datasets: Caller (where testing functions are rolled back) and Callee (where imported dependencies are rolled back). The paper evaluates various LLMs (LLaMA, DeepSeek, GPT, Claude) with OpenHands scaffold on their ability to recover from out-of-sync states through independent exploration and collaborative assistance. Authors identified a few interesting findings: 1) substantial performance gaps among agents (Claude-3.5-Sonnet performing best at 28.18%), 2) consistently low collaboration willingness across all agents (≤4.86%), 3) positive correlation between collaboration and recovery success, and 4) limited resource awareness among agents when facing time and budget constraints.

**Claims And Evidence:**

The claims made in the submission are generally well-supported by the evidence presented. While the paper has been a good read, there exists a few limitations that undermine the substantiality of the contribution:

1. While the paper evaluates seven different LLM agents, it employs OpenHands as the sole agentic backbone. It remains unclear whether the observed patterns (e.g. collaboration willingness and resource sensitivity) ) might be influenced by the specific interaction capabilities of this agentic framework. To this end, I'd strongly recommend authors adding at least one more agent to strengthen the generalizability of these conclusions.

2. The benchmark, though substantial, is restricted to Python repositories, representing only a portion of the software development landscape. Additionally, the synthetic nature of the out-of-sync scenario creation through git history rollback raises questions about how well these scenarios capture the complexities and nuances of real-world collaboration contexts. Further validation with naturally occurring out-of-sync cases would bolster the ecological validity of the findings.

3. The paper didn't really have any evidence where the collaborator is a real human. This would be critical to capture the dynamics of human-AI collaboration as currently it is more of AI-AI collaboration.

**Essential References Not Discussed:**

The authors should discuss relevant past work in collaborative software engineering e.g. https://arxiv.org/pdf/2406.11912. The authors should also discuss collaborative agents (in the general domain) as a general topic of interest.

**Experimental Designs Or Analyses:**

Please see comments in `Claims And Evidence` and `Methods And Evaluation Criteria`. For minor comments please refer to `Questions For Authors`.

**Methods And Evaluation Criteria:**

The proposed methods and evaluation criteria are well-designed for the problem at hand. The benchmark curation is still synthetic (see my comment above) and could be improved in the future.

I would also suggest authors adding an explicit cost metrics – this is not limited to the cost of LLMs but also the number of builds (which can take long for larger repos) and the time consumed overall. This would help make the story complete.

**Other Comments Or Suggestions:**

1. The results show that more challenging tasks benefit more from collaborative assistance. Have you explored whether there's a threshold of task complexity beyond which collaboration becomes essential for successful recovery?

2. Authors simulate collaborators using LLMs. How well do you think these simulated collaborators represent human developers in terms of their feedback and assistance patterns? Have you considered validating this approach with real human collaborators on a subset of tasks?

3. at line 112, why in `Update State T1` leads to $S_1 \neq B_1$?

4. Looking at the prompt it seems the cost has not been quantified correctly, thus it makes sense that agent has "low sensitivity to financial resources". Could you explicitly define what costs mean here (e.g., time, money for LLM call, builds) and see whether the conclusion still stands?

5. at line 363, how about those communications that result in success but uses more turns compared to no communication?

6. Many figures especially Figure 2 and 6 are very difficult to read. In addition, the use of colors throughout the paper is quite distractive especially for certain group of people.

**Other Strengths And Weaknesses:**

See above sections especially `Claims And Evidence` and `Methods And Evaluation Criteria`.

**Questions For Authors:**

See `Other Comments Or Suggestions`

**Relation To Broader Scientific Literature:**

The paper was situated at the intersection of code agents and collaborative systems in software development.

**Theoretical Claims:**

N/A

---

> ### Author Rebuttal · Authors · 2025-04-01
>
> Thank you for your valuable comments. We are honored that you find our work to be well-designed. Your kind suggestions, such as backbone diversity, related work, human validation, and figure settings, also provide constructive insights that we would take into consideration in our revision.
>
> 1. We appreciate your suggestions on different platforms. Comparing seven LLM agents, we use OpenHands for more controlled Env interaction. We will also apply our methods to other platforms to strengthen our work.
>
> 2. In regard to data source, generalizability, and practical use, we define agent out-of-sync to be language-agnostic. While considering the differences among diverse languages, we hope to focus on one language at each time with sufficient data to better evaluate and enhance LLM agents’ out-of-sync recovery abilities. Constructing Caller and Callee for complexity hierarchy, we will make further exploration based on real-world nuances and future improvements discussed in Sec A. Making our methods (adaptable to different repositories and languages) open-source to help the community expand and customize SyncBench, we will also enrich SyncBench with out-of-sync tasks from real-world issues and PRs and repositories with different primary languages, complexity, and domains.
>
> 3. Thank you for your comments on cost metrics. As our resource-aware recovery (Sec 2.3) includes collaborator assistance (CA) costs and solution validation (SV) costs for resource consumptions of both collaborators and executions, we will further break down costs into sub-metrics based on task-specific build time and codebase scale. We would also like to enrich your suggestions with system capacity that can hugely affect the builds and time of the same repository.
>
> 4. Reply to Questions
>
> (1) Tab 1 reveals that agents tackling more challenging tasks generally benefit more from CA, whose extent of impact depends hugely on agents’ willingness to collaborate, their reasoning and coding abilities, and how well they understand and utilize collaborators’ responses, besides task complexity. Pinpointing a precise task complexity threshold is therefore challenging, as it combines with other factors to exert effects, and varies significantly across different models and task settings (e.g., DeepSeek with the least willingness to collaborate benefits much less from CA than Llama-3.1-8B with the lowest $SR$; Claude-3.5-Sonnet with the highest $SR$ gains less than GPT-4o that is better trained for interactive use).
>
> (2) Thank you for your insights into human-AI collaboration. Aiming to tackle the challenge of agent out-of-sync in general, we design know-everything agents as either human or AI collaborators (Sec 3.3, 4.4, D.5). Supported by recent work with LLMs as humans, we further validate our design via upper bound experiments, whose performance ($SR=86.33$%) demonstrates the effectiveness of not only LLM-simulated human/AI agents, but know-everything collaborators in providing useful feedback and assistance. We will also include human-in-the-loop experiments to better support human-AI collaborations.
>
> (3) Initially $S0=B0$ at $T0$, collaborator update at $T1$ changes the repository state, resulting in $S0 \neq S1$. However, the agent is unaware of the update at $T1$, so its belief state is unchanged from $T0$ to $T1$, lending to $B0=B1$. Consequently, we have $S1 \neq B1$.
>
> (4) Allow us to extend our response from 3: Our resource-aware recovery implements costs with an initial budget (i.e., agent’s discretionary budget at the beginning of the task), CA cost (i.e., to quantify the time and cost consumed by the collaborator to gather information and assist the agent along with the time and LLM call cost of the agent), and SV cost (i.e., to quantify the time and cost of the agent to reach current state and propose the solution, along with the time and cost, e.g., build testing env and execute tests, to evaluate agent’s solution on five metrics).
>
> (5) This is one of the cases occurring due to the agent's limited communication abilities, especially in raising high-quality questions (Sec C.4, Tab C4), while it is hard to decide which is more resource-efficient (especially considering both time and cost) given different action trajectories: e.g., 20-turn success with 0 CA and 10 SV may cost more than 25-turn success with 5 CA and 1 SV. We therefore assess question quality based on final success. Experiments also reveal agents’ limitations in not only collaboration initiative, but communication quality and strategy (Sec 4.4, 4.5, C.4): e.g., if an agent first asks collaborator the localization of the update $U$ at $T1$, it can save many turns of independent Env exploration to localize and understand $U$. We aim to enhance agents’ collaboration awareness and communication capabilities with more sophisticated metrics design in future work.
>
> (6) Huge thanks for kindly suggesting figure settings. We will improve our figures in both our revision and our future work.

---

### Official Review · Reviewer_ERFr · 2025-03-19

**Overall Recommendation:** 3

**Summary:**

The paper tackles the challenge of out-of-sync collaboration, where an automated agent powered by LLM encounters errors due to a state change of the underlying codebase. The primary contributions of the paper are SyncMind, a framework for defining, identifying, and evaluating such issues, and SyncBench, a benchmark of out-of-sync scenarios derived from real codebase histories from GitHub.
The paper accounts for multiple modes of recovery, such as independent recovery - through interacting with the environment and proposing solutions, and collaborative recovery - through collaborator assistance. The latter could be a human or an agent but is portrayed by an agent for the experiments. The paper defines five metrics to measure recovery performance. Success Rate (SR) measured by execution test and parsing validation, Localization Accuracy (LA), which can be measured at the file or function level for correctness, with the function being a broadened term used including methods, Conditional Success Rate (CSR) which measures recovery conditioned on localization, Assistance Seeking Rate (ASR), and recovery efficiency. The last of which is an interesting metric as they find that most agents have poor resource awareness, and are not able to flexibly adjust plans based on the available compute. Other interesting finds of the study include the comparisons of independent vs collaborative results, where collaboration generally improves the performance, though there seem to be a large number of cases of adverse effect as well. And the low willingness of the models to cooperate. The testing is done on 300 instances of SyncBench, downsampled from 24,332.

**Claims And Evidence:**

Most of the claims made in the paper are supported by evidence.
Some claims that could benefit from further evidence are

- Agents with More High-Quality Questions Achieve Better Performance - While fig. 7 seems to support this, the definition of "high quality" where low is questions resulting in recovery failure or high resulting in recovery success, seems to be correlated with the performance.
- Collaboration improves recovery success -while there does generally appear to be the trend acrroding to fig. 6 and tab. 1, there are enough cases of no or negative impact to warrant further investigation. The magnitude of improvement also varies considerably.
- Real-world similarity of SyncBench, while the filters make sense, the automated pipeline still seems like it could contain out-of-scope scenarios.
- Low collaboration willingness - Is supported by the results, though there does not seem to be a dedicated incentive structure to push LLMs towards this, and considering high malleability of LLMs to a given task it is reasonable to expect significant differences in behaviour is such is applied.

**Essential References Not Discussed:**

The paper provides a reasonable coverage of prior works.

**Experimental Designs Or Analyses:**

I have not checked the details of the experiments such as code, and I believe the original dataset was not provided. However, the setup as described makes sense, especially for the execution is solid.

**Methods And Evaluation Criteria:**

The paper has a benchmark and evaluation metrics as primary contributions. The dataset is based on 21 repositories, which might somewhat limit its diversity. While the paper claims the framework to be easily extendable, it raises some concerns about the generalizability of the current results. The multi-level filtering for the commit seems reasonable, though it is not clear whether the remaining cases would all be related to out-of-sync issues or some other causes as well. The execution setup itself, supported by dedicated Docker environments is a great choice.

The metrics chosen also make sense, with SR and LA being intuitive choice, and the addition metrics like CSR, ASR, and Recovery Efficiency providing interesting insights into the agent performance.

The mode choices present a reasonable sampling across open (LLaMA family, DeepSeek) and close (GPT family, Claude), small (LLaMA 8B, gpt-40 mini) and larger (LLaMA 70B, GPT4-O) models.

**Other Comments Or Suggestions:**

NA

**Other Strengths And Weaknesses:**

The paper tackles an important and novel problem and provides a comprehensive benchmark with detailed evaluation. It also reports results for some of the most popular models, highlighting the challenges in out-of-sync collaboration recovery in the current generation of LLMs. The paper is well-written and easy to follow. Incorporating the efficiency analysis and the scale of the model's inability to accurately budget are very interesting findings.

Some weaknesses include:
- While the 86.33 upper bound for collaboration with oracle is interesting, the performance of the all-seing collaborator model is underexplored.
- Relatively limited github repositories used and the singular (Python) language which limits generalizability of the findings.
- Lack of a human in the loop, as those would be representative of most real-world use cases.
- The quality of the samples after the multi-stage filtering as discussed earlier.

**Questions For Authors:**

How likely are the cases in SyncBench to occur in a project that uses programming best practices for version control and continuous testing?

What is the variance in performance within the model itself for collaborative versus Independent scenarios? Do the models that gain or lose from collaboration consistently do so?

Would you expect out-of-sync issues identified to be generalizable to other languages?

**Relation To Broader Scientific Literature:**

There has been a large amount of work in getting LLM agents more effective in real-world coding settings [1,2]. The paper highlights an understudied issue – dynamic environments in collaborative coding – and provides a comprehensive evaluation framework. It builds upon prior work that primarily focuses on static environments. It also relates and provides a coding-based benchmark to build upon existing works on Theory of Mind (ToM) in LLMs [3], framing the out-of-sync problem as a failure of the agent to understand the current state of the codebase. The paper implicitly connects to research on multi-agent systems and resource-bounded reasoning, and could be followed up in that direction.

1. Liang, J. T., Yang, C., & Myers, B. A. (2024, February). A large-scale survey on the usability of ai programming assistants: Successes and challenges. In Proceedings of the 46th IEEE/ACM international conference on software engineering (pp. 1-13).
2. Jiang, J., Wang, F., Shen, J., Kim, S., & Kim, S. (2024). A survey on large language models for code generation. arXiv preprint arXiv:2406.00515.
3. Chen, Z., Wu, J., Zhou, J., Wen, B., Bi, G., Jiang, G., ... & Huang, M. (2024). Tombench: Benchmarking theory of mind in large language models. arXiv preprint arXiv:2402.15052.

**Theoretical Claims:**

NA

---

> ### Author Rebuttal · Authors · 2025-04-01
>
> Thank you for your valuable review. We are honored that you find our work to be novel and well-written. Your kind comments on generalizability and human-in-the-loop experiments also provide constructive insights that we would like to include in our revision.
>
> 1. We measure question quality based on whether questions can lead to successful outcomes (Sec 4.5, Fig 7), and extend it to three assessment aspects (specificity, timing, context integration) (Sec C.4). We therefore analyze the correlations of question quality $SR, LA_{file}, LA_{func}, ASR, CSR$, question categories, and question characteristics. Additionally, we assess and analyze question quality based on categorization and recovery effects (Sec C.4, Tab C4).
>
> 2. We conduct upper bound experiments by providing the all-seeing collaborator with complete task-specific contexts and ground truths to assist the coding agent in single-turn recovery ($ASR=100$%). Results ($SR=86.33$%, $LA=100$%) validate not only the reliability of LLM-simulated collaborators, as either humans or AI agents, in effectively providing high-quality task-specific assistance to coding agents, but the untapped potential of coding agents in proactively interacting with collaborators, efficiently obtaining and understanding relevant information of significance for current task, and effectively utilizing collaborator assistance to recover its out-of-sync state.
>
> 3. We agree that collaboration generally improves $SR$, with varying influence among cases (Sec 4). To explore further, we extend our discussion to its impact variances in Sec C, especially $LA_{file}, LA_{func}$, and performance gaps among agents, as affected by their intrinsic reasoning and coding abilities, as well as their collaboration willingness and quality. Cases with contrastive effects on $LA$ and $SR$ also suggest that accurate localization cannot guarantee recovery success, which involves multiple aspects, like agent’s technical and collaboration capabilities, their help seeking facets (e.g., ask more about solution than localization), etc.
>
> 4. Thank you for suggesting the out-of-scope issue. We leverage commits to build out-of-sync scenarios, thereby ensuring that the state mismatch of initial SyncBench samples is based on history commits without going beyond temporal out-of-sync scenarios (Sec 3, A, B.2). We then apply multi-level filtering to filter out low-quality data to better reveal insightful findings.
>
> 5. After discovering LLM agents’ unwillingness to collaborate in preliminary tests, we further push them to collaborate by adding incentive instructions in prompt to encourage them to ask for collaborator assistance: e.g., (Sec D.1) the last sentence `**Tips**...` in the input prompt to specifically encourage the agent to ask for collaborator’s assistance.
>
> 6. Reply to Questions
>
> (1) For projects with robust CI/CD and version control, semantic inconsistencies remain challenging for collaborative programming, as collaborators with different belief states need to work on individual tasks from now and then, causing temporal belief state mismatches given the dynamic collaboration environments. We therefore aim to introduce SyncBench and SyncMind to help tackle semantic out-of-sync beyond cases that can be resolved by version control and continual testing systems.
>
> (2) In terms of each model’s performance variation, we summarize individual performance in both settings (Tab 1, C1, Fig 6) and find the general positive influence of collaborator assistance on recovery success ($SR$), with varying effects on different metrics and LLMs due to different LLMs’ intrinsic coding and reasoning capabilities. Our pilot tests (Sec B.1) and experiments (Sec 4) shows that performance variance within each model is generally consistent under the same setting at different data scales, with pilot tests furnishing a preliminary validation to determine proper experiment settings. Largely affected by models’ intrinsic abilities, we aim to reveal both general and model-specific strengths and weaknesses of different agents, providing insights into future development of human-agent and multi-agent systems.
>
> (3) Allow us to reply to both your question and earlier comments (Sec 3, A, B): Although more than many existing SE benchmarks, e.g., SWE-Bench that leverages 12 repositories, we agree that 21 repositories (all with multiple languages, primarily Python) may still limit generalizability. Yes, we define agent out-of-sync to be language-agnostic. Considering the differences among diverse languages, we hope to focus on one language at each time with sufficient data to better evaluate and improve LLM agents’ out-of-sync recovery abilities in each language. We will also include more repositories with different primary languages, complexity, and domains, into SyncBench, and make our construction method (adaptable to different repositories and languages) open-source to help the community enrich and customize SyncBench further.

---

### Decision · Program_Chairs · 2025-05-01

**Decision:**

Accept (poster)

**Comment:**

**Meta-Review of “SyncMind: Measuring Agent Out-of-Sync Recovery in Collaborative Software Engineering”**

**Summary of the Paper**
This paper addresses the challenge of AI-powered agents losing sync with a dynamically changing codebase in collaborative software engineering (CSE). The authors introduce **SyncMind**, a framework for systematically defining and studying “out-of-sync” scenarios—when an agent’s belief about the code base diverges from the current repository state. Built on SyncMind, the authors propose **SyncBench**, a benchmark containing 24,332 out-of-sync instances from 21 real-world GitHub repositories with executable tests. In experiments involving multiple large language model (LLM) agents (e.g., Claude, GPT-4o, Llama variants), the paper measures agent performance on (1) out-of-sync recovery success, (2) collaboration willingness (whether the agent actively seeks help), and (3) resource awareness (time/financial budgeting). Findings suggest that while collaboration and extra time can improve performance, LLM agents show surprisingly low initiative to collaborate and limited adaptation to resource constraints.

---

## Strengths

1. **Out-of-sync Problem**
   - The paper defines *out-of-sync* state in CSE, focusing on the mismatch between an agent’s internal belief and the up-to-date code base. This extends beyond benchmarks that assume a static repository and handles a realistic scenario in which collaborator commits may desynchronize the agent’s mental model of the code base.

2. **SyncBench Benchmark**
   - SyncBench is drawn from commit histories of popular repositories. The instances are systematically filtered and verified to ensure that each scenario captures a transition from working code to broken code due to an outdated function or dependency.
   - The benchmark is also *large* in scale (over 24k out-of-sync instances) and diverse in coverage even though the final set used in experiments was downsampled to 300 for cost reasons.

3. **Evaluation across Multiple Metrics**
   - The authors propose five metrics (Success Rate, Localization Accuracy at file/function levels, Conditional Success Rate, Assistance Seeking Rate, and Recovery Efficiency) evaluating both technical correctness and collaborative behaviors.
   - The paper investigates *resource awareness* (time and financial budget), an important dimension that is sometimes neglected in AI-for-SE evaluations.

4. **Evaluation Results**
   - Evaluation uses seven LLMs and shows varied performance in identifying out-of-sync code, localizing failures, and fixing them.
   - Collaboration typically improves success rates but is rarely invoked; even high-performing models exhibit low willingness to ask for help.
   - Resource constraints (time and cost) have minimal effect on behavior, highlighting that current LLM agents do not adapt well to usage constraints, an important real-world consideration.

5. **Well-Written, Accessible Presentation**
   - The authors supply extensive experimentation details.

---

## Main Weaknesses and Concerns

1. **Single Language and Platform**
   - **Single-Language**: The benchmark is built from Python repos. There is no analysis of how well this would apply to other programming languages.
   - **One Agentic Platform (OpenHands)**: The environment uses a specific open-source agent framework. There is no analysis for other agent architectures.
   - Additional validation or expansions to more varied repositories (front-end, multi-language, large monolithic code bases, etc.) would strengthen the benchmark.

2. **Absence of Real Human–AI Collaboration Experiments**
   - While the authors simulate collaboration with a “know-everything” LLM-based collaborator, real human interactions can reveal unique behaviors (e.g., partial hints, style differences, trust or frustration issues). Multiple reviewers remarked that human-in-the-loop experiments would improve the research results.

3. **Concerns About Data Realism**
   - The out-of-sync scenarios are generated by rolling back code from commit history. Although this does create real pass→fail transitions, some reviewers worry that “offline” history-based data might not fully capture large-scale multi-developer concurrency or intricacies of real live collaboration.

---

## Recommendation

All three reviewers see merit in the core problem definition, the benchmark, and the systematic evaluations. Most critiques revolve around extending the scope (more repositories/languages, real human collaborators, and further resource modeling).

Based on the reviewers' evaluation, I recommend accepting this paper.